# Postoperative management of total elbow arthroplasty: Results of a European survey among orthopedic surgeons

**Willemijn van Dam**[1◉]\*, **Danielle Meijering**[1◉], **Martin Stevens**[1◉], **Alexander L. Boerboom**[1◉], **Denise Eygendaal**[2◉]

1 Department of Orthopedic Surgery, University Medical Center Groningen, University of Groningen, Groningen, The Netherlands, 2 Department of Orthopedics and Sports Medicine, Erasmus Medical Center, Rotterdam, The Netherlands

◉ These authors contributed equally to this work.
\* w.van.dam@student.rug.nl

## Abstract

### Background

The number of complications after total elbow arthroplasty (TEA) is high and survival rates are low compared to hip and knee arthroplasties. The most common reason for revision is aseptic loosening, which might be caused by overloading of the elbow. In an attempt to lower failure rates, current clinical practice is to restrict activities for patients with a TEA. However, postoperative management of TEA is a poorly investigated topic, as no evidence-based clinical guidelines exist and the aftercare is often surgeon-based. In this study we evaluated the current postoperative management of TEA among orthopedic surgeons.

### Methods

An online survey of 30 questions was sent to 635 members of the European Society for Surgery of the Shoulder and the Elbow (SECEC/ESSSE), about 10% (n = ± 64) of whom are considered dedicated elbow specialists. The questions were on characteristics of the surgeon and on the surgeon's preferred postoperative management, including items to be assessed on length of immobilization, amount of weight bearing and axial loading, instructions on lifelong activities, physiotherapy, and postoperative evaluation of the elbow.

### Results

The survey was completed by 54 dedicated elbow specialists from 17 different countries. Postoperative immobilization of the elbow was advised by half of respondents when using the triceps-sparing approach (52%), and even more with the triceps-detaching approach (65%). Postoperative passive movement of the elbow was allowed in the triceps-sparing approach (91%) and in the triceps-detaching approach (87%). Most respondents gave recommendations on weight bearing (91%) or axial loading (76%) by the affected elbow, but the specification shows significant variation.

**Data Availability Statement:** All relevant data are within the manuscript and its Supporting Information files.

**Funding:** The authors received no specific funding for this work.

**Competing interests:** The authors have declared that no competing interests exist.

## Conclusion

The results from this survey demonstrate a wide variation in postoperative care of TEA. The lack of consensus in combination with low survival rates stresses the need for clinical guidelines. Further research should focus on creating these guidelines to improve follow-up care for TEA.

## Introduction

Global incidence of total elbow arthroplasty (TEA) varies between 5 and 7.5 per 1000 inhabitants [1]. Numbers of TEAs are growing, and almost doubled between 1998 and 2011 in the United States [2]. In the Netherlands the numbers of TEAs have risen too, from 67 in 2017 to 79 in 2019. These figures are rather low compared to yearly hip (n = 25,500) and knee (n = 20,200) arthroplasties [3]. Indications for TEA have changed over the years, from rheumatoid arthritis and osteoarthritis to posttraumatic arthritis and acute fractures. The postoperative management of TEA has not been widely discussed in the current literature, and no standardized post-TEA protocol exists. It is therefore hypothesized that this leads to variable and inconsistent postoperative care.

Although results of TEA have improved over the last years, survival rates are still relatively low (79.2%) and complication rates remain high (11–38%) compared to hip or knee arthroplasty [4–6]. Aseptic loosening constitutes one of the main reasons for prosthetic failure (38%) and is most likely caused by overloading of the elbow joint [7]. In an attempt to decrease overall failure, common practice is to restrict elbow activities following TEA [8].

A major source of the absence of clinical guidelines is the dearth of studies that contribute to postoperative care in terms of which elbow activities may be allowed and following what timeline: some institutions advise avoiding weight bearing by the affected elbow [9, 10], while others seem to focus more on restricting the range of motion [11]. To that end, the objective of this study is to create insight into the current postoperative care of TEA, aiming to improve treatment by means of a questionnaire-based survey among specialized orthopedic surgeons.

## Methods

### Procedure

An online survey of 30 questions was sent to 635 members of the European Society for Surgery of the Shoulder and the Elbow (SECEC/ESSSE). The questionnaire was addressed specifically to dedicated elbow specialists. The web-based survey tool "google forms" was used. In November 2020 the questionnaire was disseminated by SECEC via email invitation with a link to the survey. In the invitation it was explained that completion of the survey was taken as consent to participate. Reminder emails were sent two and four weeks following the first invitation, to increase response rate. Data collection was closed December 7, five weeks after the first email was sent. The study was conducted in accordance with the Dutch law (WMO) and regulations of the local medical ethical committee. As no patients were involved in the study there was no need for approval from the Medical Ethics Committee of University Medical Center Groningen (UMCG).

### Questionnaire

The questionnaire consisted of 30 questions about different topics: general information on demographics and respondents' experience; information on surgeons' most commonly used

surgical approach and prosthetic design; recommendations for immobilization, movement (e.g., active or passive), specific instructions concerning lifelong activities (e.g., axial loading and weight bearing), and physiotherapy; differences in postoperative advice related to the surgical approach and indication for surgery; and questions on how and when postoperative assessment of the elbow occurred and whether triceps function was being monitored. In the questionnaire a distinction between triceps-sparing and triceps-detaching (e.g., triceps-flap, -reflecting, -splitting) surgical approaches was made according to a classification described by Booker and Smith [12]. The questionnaire was developed by our study group and reviewed by two Dutch experts (ALB and DE) in the field of TEA (S1 Appendix).

### Statistical analyses

Statistical analyses were performed using IBM SPSS Statistics (version 23.0, Chicago). Results were analyzed using descriptive statistics. Categorical variables are displayed as count (n) and percentage (%). Continuous variables with normal distribution or abnormal distribution are displayed by mean ± standard deviation (SD) or median with range, respectively.

## Results

### Respondents

A total of 54 orthopedic surgeons from 17 different countries and member of SECEC/ESSSE filled out the survey. The SECEC consists of shoulder and elbow specialists. Of the 635 members of the SECEC, about 10% (n = ± 64) is considered dedicated elbow specialist. Most respondents had up to ten years of experience (54%), and the majority performed ten or fewer TEAs annually (76%) (Table 1). The Latitude EV (® Stryker/Wright Medical) was the most commonly used prosthesis (44%), followed by the Coonrad-Morrey (® Zimmer Biomet) (27%) and the Nexel (® Zimmer Biomet) (20%) prostheses. The triceps-sparing surgical approach was most commonly used (59%), followed by the triceps-splitting (18%), posterior triceps-flap (16%), and triceps-reflecting approaches (7%).

Non-response percentages varied for questions that weren't applicable to the surgeon. Results percentages will therefore be described by the distribution of the completed answers.

### Postoperative immobilization of the elbow

Of the 33 orthopedic surgeons using the triceps-sparing approach, 17 (52%) advised immobilization of the elbow. Of the 23 orthopedic surgeons using the triceps-detaching (e.g., triceps-splitting, -flap and -reflecting) approaches, 15 (65%) advised immobilization of the elbow (Fig 1).

A total of 33 respondents gave recommendations on the length of elbow immobilization, regardless of surgical approach used. The total number of days that a removable cast should be

**Table 1. Descriptives of respondents (n = 54).**

| Variable | Value | Respondents |
|---|---|---|
| Years of experience | 0–10 | 29 |
| | 11–20 | 15 |
| | > 20 | 10 |
| Annually performed TEAs | 0–10 | 42 |
| | 11–20 | 7 |
| | > 20 | 5 |

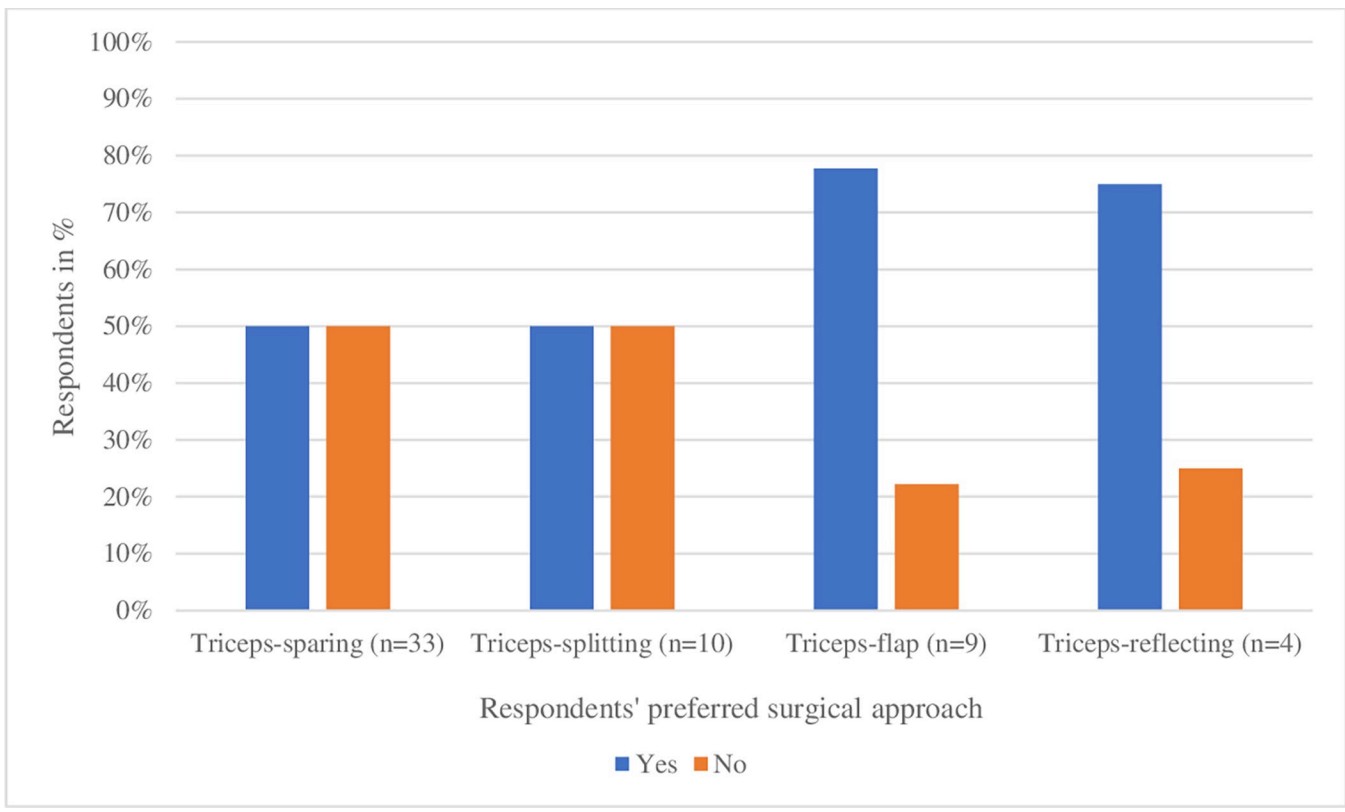

**Fig 1. Recommended postoperative immobilization of the elbow by respondents, specified per surgical approach (n = 54).** Two respondents gave recommendations on two different approaches, while 52 respondents described just one approach, adding up to 56 answers.

worn was 1–5 days advised by eight out of 33 respondents (24%), 6–10 days by nine respondents (27%), 11–15 days by eleven respondents (33%), and > 15 days by five respondents (15%); two respondents (6%) advised immobilizing the elbow beyond 40 days.

The degrees of flexion at which elbow immobilization was advised showed variation too. Answers ranged from 0–30˚ flexion (13 respondents) to 30–60˚ flexion (6 respondents) and 60–90˚ flexion (13 respondents) (Fig 2).

## Allowed postoperative passive movement of the elbow

Of the 54 respondents, 48 (89%) allowed postoperative passive movement of the elbow: 30 respondents (91%) using a triceps-sparing approach (n = 33) and 20 respondents (87%) using the triceps-detaching approaches (n = 23). Further specification of allowed passive movement per approach is described in Fig 3. Passive movement of the elbow was most often allowed from day 1 postoperatively by 20 out of 52 respondents (39%), and from day 2 by ten respondents (19%).

## Allowed postoperative active movement of the elbow

Forty-six out of 54 respondents (85%) allowed postoperative active movement of the elbow as well. Specification of allowed active movement per approach is presented in Fig 4. On what day active movement of the elbow was allowed, varied widely (median: day 3, range: 1–42 days postoperatively) and was sometimes dependent on the occurrence of clinical signs (e.g., movement allowed by pain) or on the quality of the triceps.

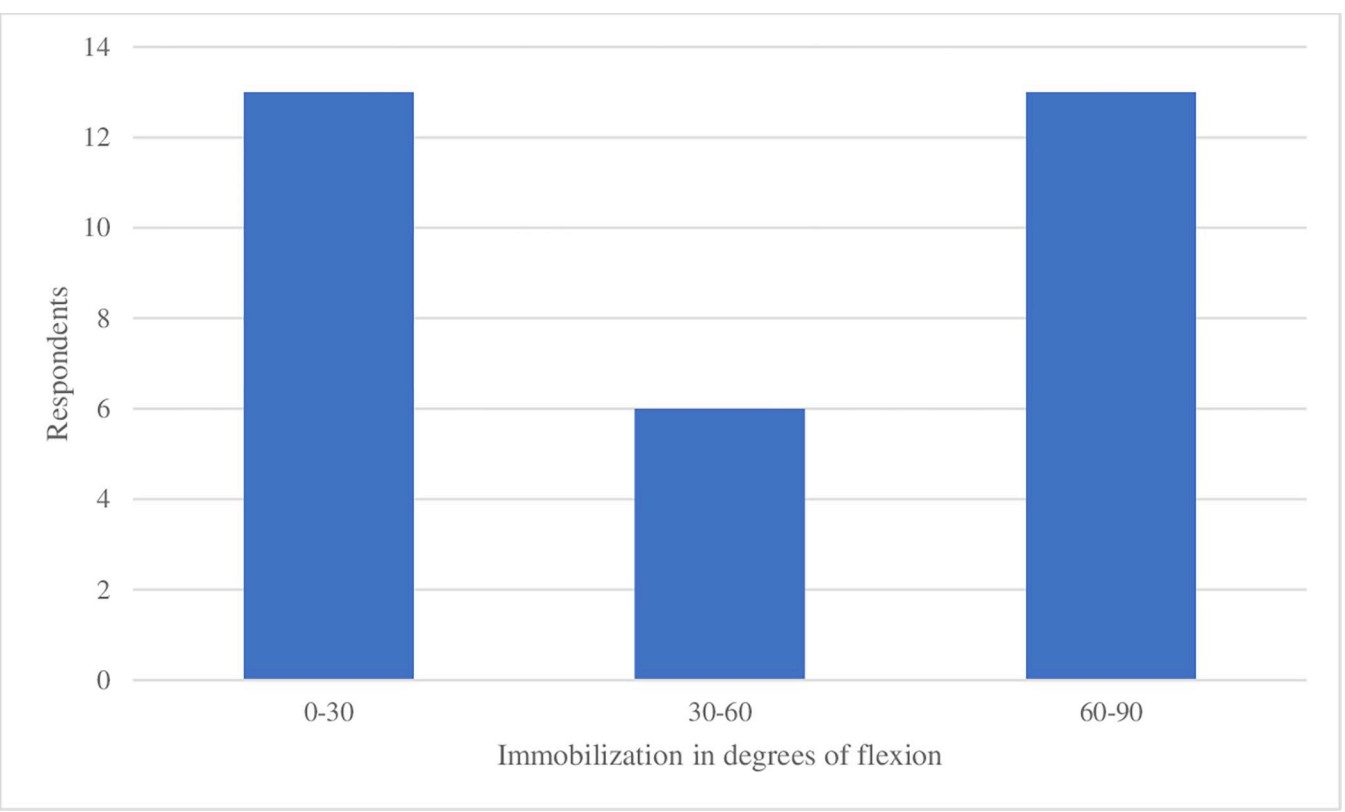

**Fig 2. Recommended position of immobilization of the elbow (n = 32).**

### Instructions on weight bearing and axial loading

In general, respondents were very protective in their recommendations on mechanical loading of the elbow. Forty-nine out of 54 respondents gave recommendations on weight bearing by the elbow. A maximum amount of 1–5 kg weight bearing was advised by 41 respondents (84%), and < 1 kg by five respondents (10%).

Forty-one out of 54 respondents gave recommendations on axial loading by the elbow. The maximum amount of 1–5 kg axial loading was advised by 30 out of 41 respondents (73%), < 1 kg by six respondents (15%), and 1–10 kg by four respondents (10%) (Fig 5).

### Lifelong instruction on activities

Slightly over half of respondents (n = 30, 56%) gave specific lifelong instructions about certain activities. Eighteen out of 30 respondents (60%) agreed on avoiding heavy physical activities with the elbow. Other instructions were *"no repetitive work"* by two respondents (7%), *"no impact sports"* (7%), *"daily life activities allowed"* (7%), and *"avoid rotational forces"* (3%).

### Supervision by physiotherapists

Postoperative supervision by a physiotherapist was advised by 49 out of 54 respondents (91%) but the duration of physiotherapy varied widely, ranging from 3 weeks to 1 year.

### Postoperative assessment of the elbow

Postoperative assessment of the elbow was performed in an outpatient clinic by 51 of 54 respondents (94%), again with a wide variety of time intervals. Only four respondents (7%)

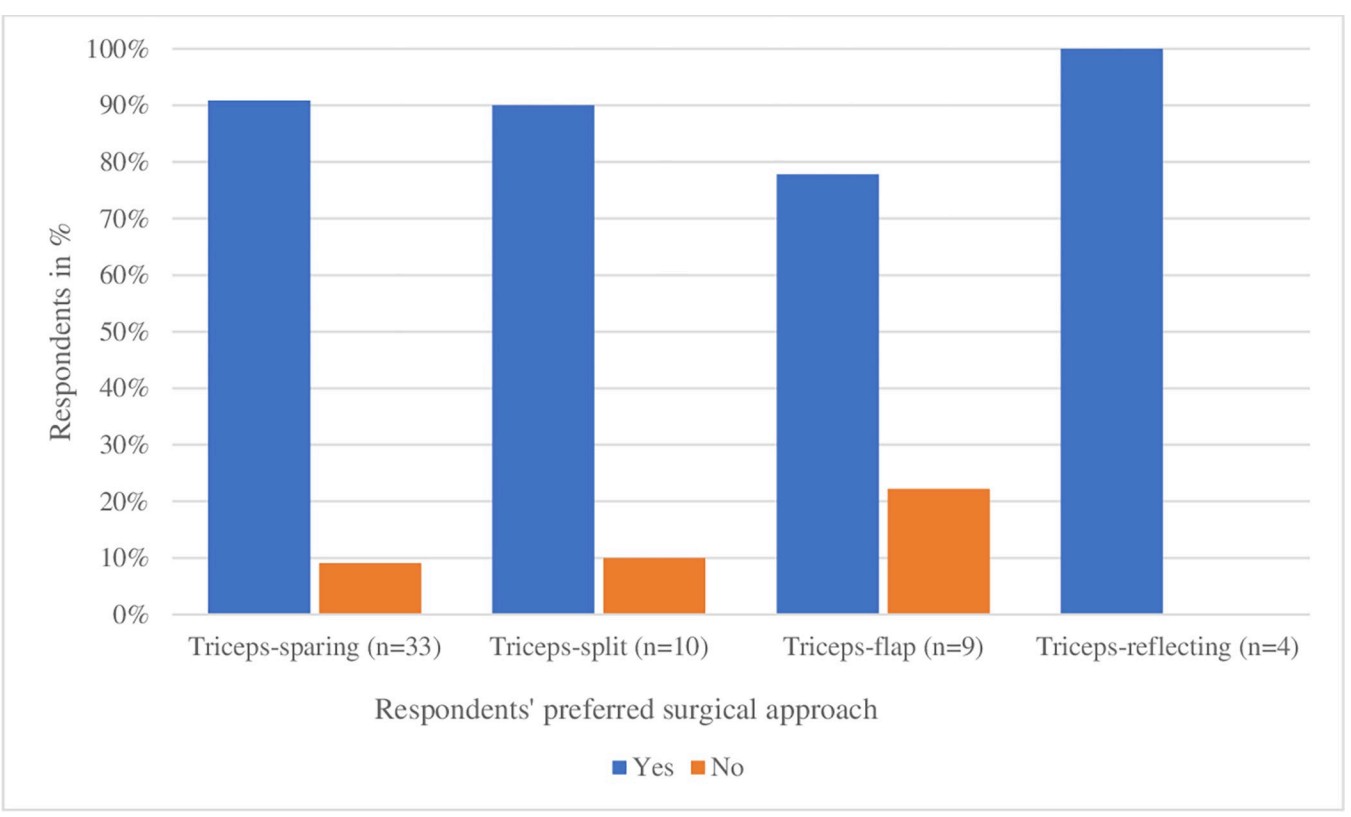

**Fig 3. Passive movement of the elbow allowed postoperatively by respondents, specified per surgical approach.** Two respondents gave recommendations on two different approaches while 52 respondents described just one approach, adding up to 56 answers.

performed postoperative assessment of the elbow at four weeks, while 36 respondents (67%) did so at twelve weeks, among other moments (Fig 6).

## Postoperative monitoring of the triceps muscle

Triceps function following TEA was monitored by 41 out of 54 respondents (76%). Most respondents (n = 33, 80%) monitored the triceps by testing active extension of the elbow against gravity, others used the Medical Research Council Scale for muscle strength (MRC scale) (17%). Only one respondent (n = 1) used a *"special triceps power measurement device"* to determine triceps function.

## Differences in postoperative protocol according to indication, primary or revision

Thirty-seven out of 54 respondents (69%) stated that their postoperative protocol did not depend on the indication for TEA. For 26 out of 54 respondents (48%), postoperative protocol for TEA showed differences for primary or revision TEA, while for the others this did not make any difference. For 14 out of 26 respondents (54%) the recommended postoperative protocol following revision surgery contained more limitations in terms of movement, exercise, mechanical stress, or protection of the triceps muscle. In addition, seven out of these 26 respondents (27%) stated that their postoperative protocol for revision showed more limitations due to the difference in surgical approach (i.e., triceps-reflecting).

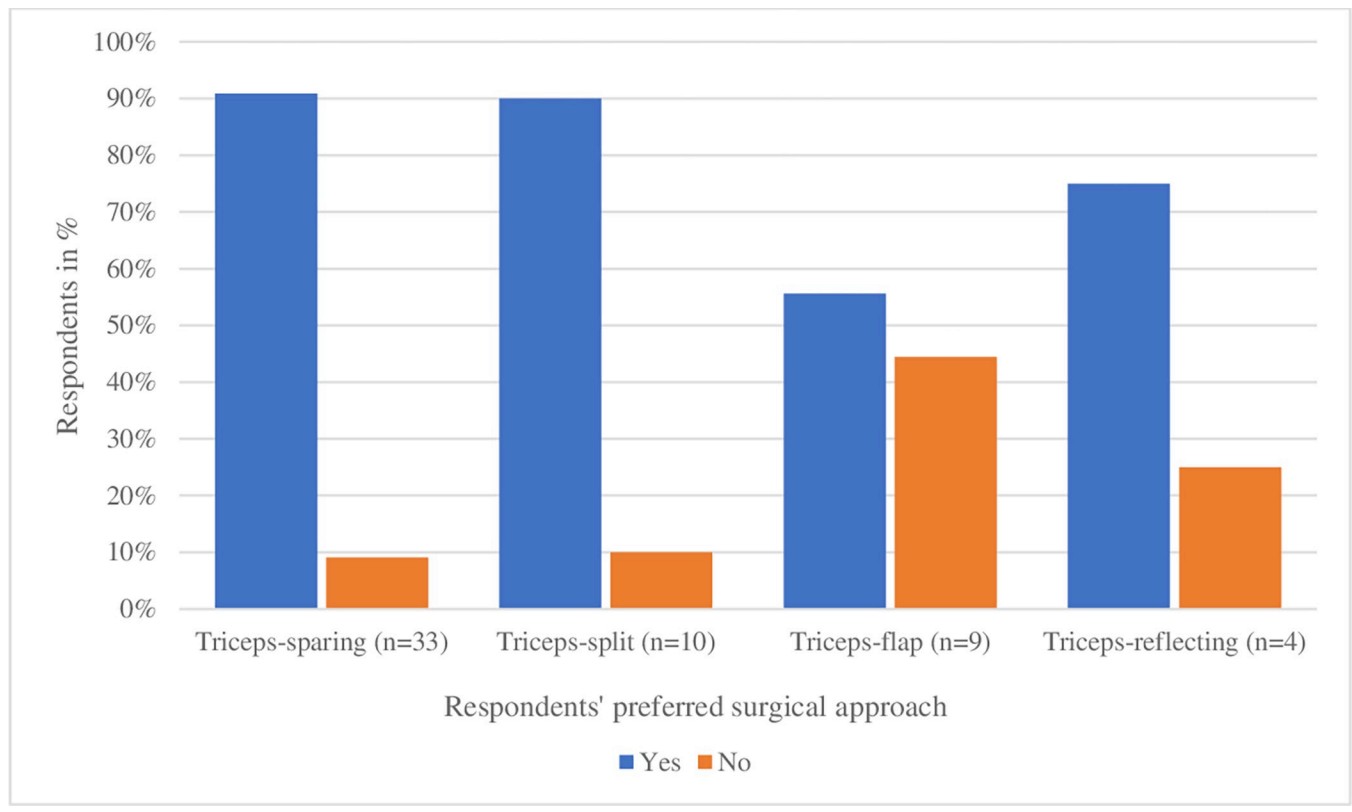

**Fig 4. Active movement of the elbow allowed postoperatively by respondents, specified per surgical approach (n = 54).** Two respondents gave recommendations on two different approaches while 52 respondents described just one approach, adding up to 56 answers.

## Discussion

In this online survey we evaluated current postoperative care of TEA among 635 SECEC/ESSSE members. As SECEC is not able to divide their members into elbow and shoulder specialists, it isn't known exactly how many of these members are dedicated elbow specialists.

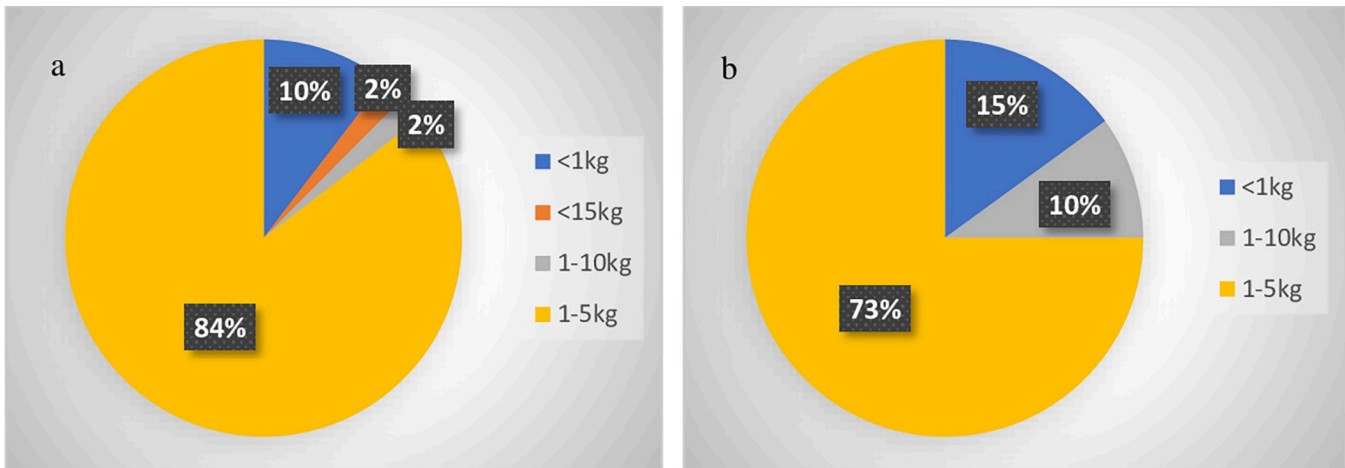

**Fig 5. Postoperative advice on elbow weight bearing and axial loading.** 5a: Advice on weight bearing by the elbow (n = 49). 5b: Advice on axial loading of the elbow (n = 41).

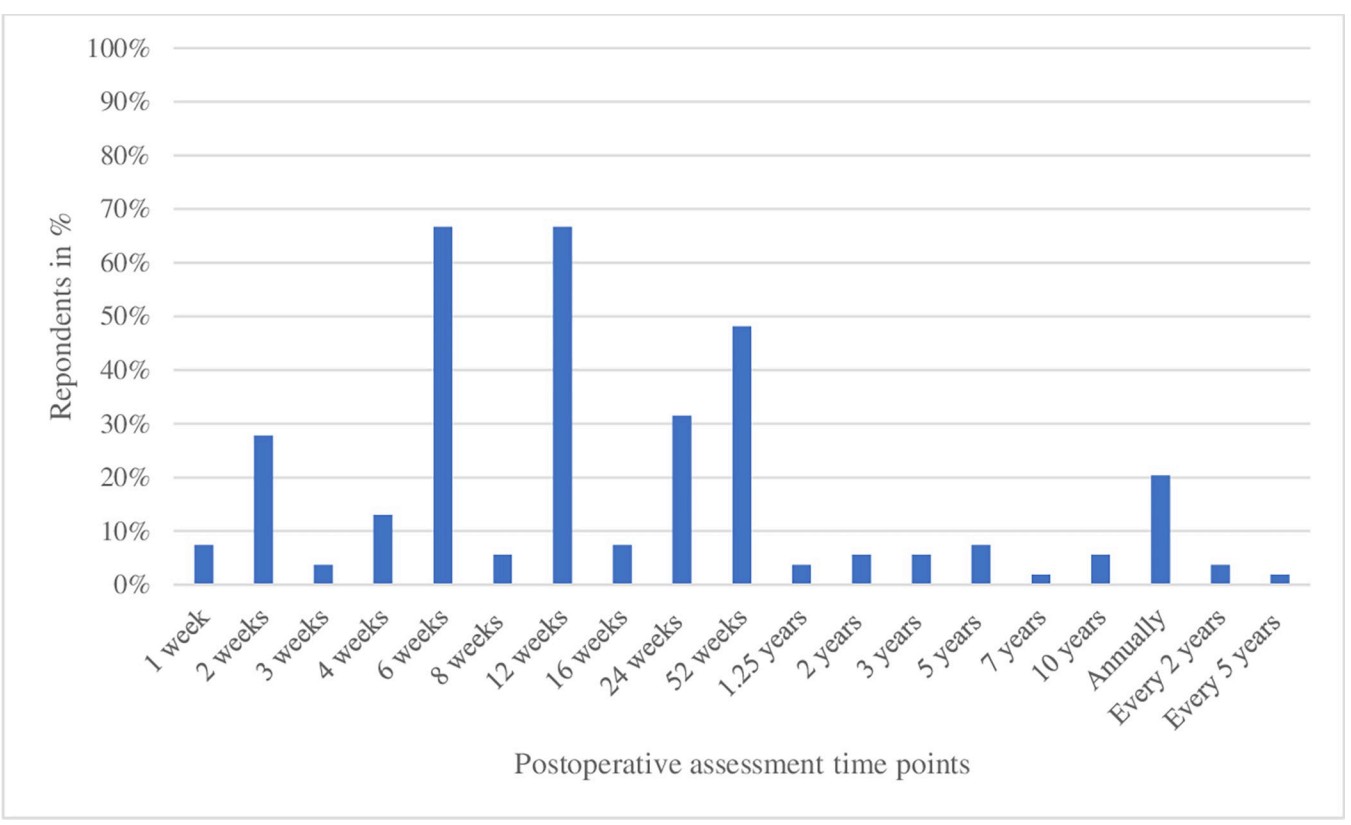

**Fig 6. Moments of postoperative assessment of the elbow.** Percentage is the number of responding orthopedic surgeons (n = 54) performing postoperative assessment of the elbow at any time. Multiple moments may be filled out per respondent.

However, based on worldwide data from national arthroplasty registries we expect about 10% (n = ± 64) to be more or less dedicated to elbow arthroplasty. We therefore believe that the response of 54 SECEC/ESSSE members is a good representation of dedicated elbow arthroplasty surgeons in Europe.

The results of this survey have made it clear that there is no consensus on a postoperative protocol for TEA. Considerable variation was found in terms of allowed movement of the elbow, specific instructions for lifelong activities, whether to be supervised by a physiotherapist, postoperative assessment of the elbow, and whether triceps function was monitored. The variability found in postoperative management among orthopedic surgeons is in line with other research that also found variation in the recommended restrictions for activities post-TEA in an attempt to improve survival [8].

Triceps insufficiency was seen in up to 4.9% of patients undergoing a TEA with a triceps-detaching approach and in 0% of patients undergoing a TEA with a triceps-sparing approach [13]. Hence the main advantage of a triceps-sparing approach is direct functional treatment of the elbow. Still, more than half of respondents (52%) immobilize the elbow postoperatively, which is contradictory unless postoperative immobilization of the elbow is advised due to soft tissue damage.

This survey reveals that no clear agreement exists on the recommended amount of elbow axial loading and weight bearing following TEA. However, there seems to be an overall opinion to keep both to a minimum (no more than 1 kg or 1–5 kg). The overall 10-year survival rate of a TEA is 81% (95% CI, 76%-86%) [14]. This relatively low survival rate is mostly caused by aseptic loosening, which is considered to be related to overloading of the prosthesis. It may

be a direct overloading of the artificial joint in terms of the polyethylene bearing or fixation of the components in the humerus and ulna, which are relatively small bones, but could also be related to malalignment and non-anatomic force distribution leading to stress shielding and bone resorption [15]. The recommendation to keep axial loading and weight bearing restricted to prevent overloading of the elbow therefore seems well-motivated. However, no evidence exists on how much axial loading or weight bearing is allowed. Beside the lack of agreement on post-operative loading and weight bearing, it is also unclear whether post-operative loading instructions are compliant with reported failure mechanisms of the prosthesis. A total elbow prosthesis seems to interfere with moderate activities of daily living, according to the University of California at Los Angeles (UCLA) activity score. What's more, 49% of patients performed at least one high-demand activity per day [8]. It seems therefore imported to focus on biomechanical studies in order to create insight in elbow joint loading during daily living and compare these loads with reported failure limits of the prosthesis, to create insight in failure mechanisms and consequently to be able to formulate post-operative instructions.

Almost half of respondents (48%) stated that their postoperative management protocol differed between primary and revision TEA. The advised postoperative protocol for revision surgery contained more limitations (54%) in terms of movement, exercise, mechanical stress, and protection of the triceps muscle, or was dependent on the difference between types of surgical approach used in revision or primary TEA (27%). The large variety in postoperative management again stresses the need for clinical guidelines.

This survey can be viewed as a first impression on the current postoperative management of TEA among orthopedic surgeons. Fifty-four elbow surgeons in 17 countries are represented in the survey and, since TEAs are only performed in few specialized clinical centers, the representation of this survey may be considered as a clear yet very heterogeneous current practice. As the number of procedures and performing surgeons are limited, we invite our colleagues to create a more generally accepted and uniform aftertreatment based on consensus and scientific evidence.

## Conclusion

The data from this survey among orthopedic surgeons on postoperative management of TEA demonstrated a wide variation in postoperative assessment, immobilization of the elbow, and differences in management between primary and revision surgery. This great variety in postoperative management underlines the lack of consensus on this topic. Low survival rates caused by loosening (e.g., probably due to overloading) combined with the lack of consensus on postoperative management confirm the need for biomechanical loading studies of the elbow during daily activities to create insight into loading patterns, limits of the prosthesis, and consequently its reasons for failure. To improve survival rates, further research is necessary to develop a clinical guideline on postoperative management.

## Supporting information

**S1 Appendix. Survey: Postoperative management of TEA.**
(DOCX)

**S1 Data. Data survey.**
(XLSX)

## Author Contributions

**Conceptualization:** Willemijn van Dam, Danielle Meijering.

**Data curation:** Willemijn van Dam, Danielle Meijering.

**Formal analysis:** Willemijn van Dam, Danielle Meijering, Martin Stevens, Alexander L. Boerboom, Denise Eygendaal.

**Methodology:** Willemijn van Dam, Danielle Meijering, Alexander L. Boerboom.

**Project administration:** Willemijn van Dam, Danielle Meijering.

**Resources:** Willemijn van Dam, Danielle Meijering.

**Supervision:** Martin Stevens, Alexander L. Boerboom.

**Validation:** Willemijn van Dam, Danielle Meijering, Martin Stevens, Alexander L. Boerboom.

**Visualization:** Willemijn van Dam, Martin Stevens, Alexander L. Boerboom.

**Writing – original draft:** Willemijn van Dam, Danielle Meijering, Martin Stevens, Alexander L. Boerboom, Denise Eygendaal.

**Writing – review & editing:** Willemijn van Dam, Danielle Meijering, Martin Stevens, Alexander L. Boerboom, Denise Eygendaal.

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
