## [Decision Letter · Decision Letter 0]

8 Aug 2022

PONE-D-22-08673Postoperative management of Total Elbow Arthroplasty: Results of a European survey among orthopedic surgeonsPLOS ONE

Dear Dr. Dam,

Thank you for submitting your manuscript to PLOS ONE. After careful consideration, we feel that it has merit but does not fully meet PLOS ONE’s publication criteria as it currently stands. Therefore, we invite you to submit a revised version of the manuscript that addresses the points raised during the review process.

 Your manuscript has been reviewed by one peer-reviewer. Their comments suggest that your manuscript could be improved upon by providing some additional detail, especially in the discussion section. The reviewer has also recommended a citation as a part of their review. We would recommend that you thoroughly evaluated the requested reference and determine whether the articles are relevant to the current study. You may feel free to disregard references with tangible relevance to the study reported in the manuscript.  Please note that we have only been able to secure a single reviewer to assess your manuscript. We are issuing a decision on your manuscript at this point to prevent further delays in the evaluation of your manuscript. Please be aware that the editor who handles your revised manuscript might find it necessary to invite additional reviewers to assess this work once the revised manuscript is submitted. However, we will aim to proceed on the basis of this single review if possible. 

We look forward to receiving your revised manuscript.

Kind regards,

Maria Elisabeth Johanna Zalm, Ph.D

Editorial Office

PLOS ONE

https://journals.plos.org/plosone/s/file?id=ba62/PLOSOne_formatting_sample_title_authors_affiliations.pdf".

Reviewers' comments:

Reviewer's Responses to Questions

**Comments to the Author**

1. Is the manuscript technically sound, and do the data support the conclusions?

Reviewer #1: Yes

2. Has the statistical analysis been performed appropriately and rigorously? 

Reviewer #1: Yes

3. Have the authors made all data underlying the findings in their manuscript fully available?

Reviewer #1: Yes

4. Is the manuscript presented in an intelligible fashion and written in standard English?

Reviewer #1: Yes

5. Review Comments to the Author

Reviewer #1: The 9% response rate seems low and the justification for why this was actually a good representation was confusing and could benefit from clarification.

The authors also mention the need for biomechanical studies in the conclusion but don't mention them throughout the paper so it might be beneficial to have a more fleshed out thought about that in the discussion and how that can relate to postop protocol standardizations since that seems to be the main focus.

Line 228 This is not really contradictory as immobilization may be for soft tissue rest. Many of these procedures are done in elderly patients with poor soft tissue envelopes or in the setting of geriatric trauma where the skin and subcutaneous tissue is very compromised.

Line 234 The authors need to explain that total elbow arthroplasty failure is not a complete mystery and finds its origin in nonanotomic force distribution, which leads to stress shielding and bone resorption over time. The violation of elbow biomechanics is outlined in an article in the Journal of Hand Surgery. Please add this reference so that the reader can understand why these fail quite reliably.

Kaufmann RA, D'Auria JL, Schneppendahl J. Total Elbow Arthroplasty: Elbow Biomechanics and Failure. J Hand Surg Am. 2019 Aug;44(8):687-692. doi: 10.1016/j.jhsa.2018.11.020. Epub 2019 Feb 13. PMID: 30770149.

6. PLOS authors have the option to publish the peer review history of their article (what does this mean?). If published, this will include your full peer review and any attached files.

Reviewer #1: No

---

## [Author Response · Author response to Decision Letter 0]

8 Sep 2022

Reviewer 1:

Comment 1

Reviewer #1: The 9% response rate seems low and the justification for why this was actually a good representation was confusing and could benefit from clarification.

Response 

We agree this may have been confusing. We edited the manuscript in the abstract (line 32), the methods (line 71-72) and results section (line 104-107) to clarify the response rate. 

Comment 2

The authors also mention the need for biomechanical studies in the conclusion but don't mention them throughout the paper so it might be beneficial to have a more fleshed out thought about that in the discussion and how that can relate to postop protocol standardizations since that seems to be the main focus.

Response 

We added an extra statement (line 243-246) explaining that insight in the amount of axial loading and weight bearing allowed by the prosthesis will lead to more justified, standardized, postoperative advise. 

Comment 3

Line 228 This is not really contradictory as immobilization may be for soft tissue rest. Many of these procedures are done in elderly patients with poor soft tissue envelopes or in the setting of geriatric trauma where the skin and subcutaneous tissue is very compromised.

Response

We agree with the reviewer. An advantage of the triceps sparing surgical approach is direct functional treatment of the elbow. However, it is in fact true that even with the triceps sparing surgical approach, postoperative immobilization may be advised due to soft tissue damage. We added a comment (line 225-227) about this in the manuscript.

Comment 4 

Line 234 The authors need to explain that total elbow arthroplasty failure is not a complete mystery and finds its origin in nonanatomic force distribution, which leads to stress shielding and bone resorption over time. The violation of elbow biomechanics is outlined in an article in the Journal of Hand Surgery. Please add this reference so that the reader can understand why these fail quite reliably.

Kaufmann RA, D'Auria JL, Schneppendahl J. Total Elbow Arthroplasty: Elbow Biomechanics and Failure. J Hand Surg Am. 2019 Aug;44(8):687-692. doi: 10.1016/j.jhsa.2018.11.020. Epub 2019 Feb 13. PMID: 30770149.

Response

An explanation considering nonanatomic force distribution and its role in the failure of the prosthesis has been added (line 234-236) with reference to the suggested paper.

---

## [Decision Letter · Decision Letter 1]

2 Nov 2022

Postoperative management of Total Elbow Arthroplasty: Results of a European survey among orthopedic surgeons

PONE-D-22-08673R1

Dear Dr. Dam,

We’re pleased to inform you that your manuscript has been judged scientifically suitable for publication and will be formally accepted for publication once it meets all outstanding technical requirements.

Kind regards,

Faizan Iqbal

Academic Editor

PLOS ONE

Additional Editor Comments (optional):

Reviewers' comments:

Reviewer's Responses to Questions

**Comments to the Author**

1. If the authors have adequately addressed your comments raised in a previous round of review and you feel that this manuscript is now acceptable for publication, you may indicate that here to bypass the “Comments to the Author” section, enter your conflict of interest statement in the “Confidential to Editor” section, and submit your "Accept" recommendation.

Reviewer #1: All comments have been addressed

Reviewer #2: All comments have been addressed

2. Is the manuscript technically sound, and do the data support the conclusions?

Reviewer #1: Yes

Reviewer #2: Yes

3. Has the statistical analysis been performed appropriately and rigorously? 

Reviewer #1: Yes

Reviewer #2: N/A

4. Have the authors made all data underlying the findings in their manuscript fully available?

Reviewer #1: Yes

Reviewer #2: Yes

5. Is the manuscript presented in an intelligible fashion and written in standard English?

Reviewer #1: (No Response)

Reviewer #2: Yes

6. Review Comments to the Author

Reviewer #1: (No Response)

Reviewer #2: This is an interesting article on survey results regarding postoperative management of total elbow arthroplasty. I congratulate the authors on addressing the discrepancies on postoperative management of this rare arthroplasty entity by surveying the SECEC surgeons.

7. PLOS authors have the option to publish the peer review history of their article (what does this mean?). If published, this will include your full peer review and any attached files.

Reviewer #1: No

Reviewer #2: No

---

## [Editor Report · Acceptance letter]

6 Nov 2022

PONE-D-22-08673R1 

Postoperative management of Total Elbow Arthroplasty: Results of a European survey among orthopedic surgeons 

Dear Dr. Dam:

I'm pleased to inform you that your manuscript has been deemed suitable for publication in PLOS ONE. Congratulations! Your manuscript is now with our production department. 

Kind regards, 

on behalf of

Dr. Faizan Iqbal 

Academic Editor

PLOS ONE